METHODS AND PROTOCOLS

# PPNet: Identifying Functional Association Networks by Phylogenetic Profiling of Prokaryotic Genomes

Yangjie Li,[a,b,c,d] Bin Ma,[a,b,c] Kexin Hua,[a,b,c] Huimin Gong,[a,b,c] Rongrong He,[a,b,c] Rui Luo,[a,b,c] Dingren Bi,[a,b,c] Rui Zhou,[a,b,c] Paul R. Langford,[e] Hui Jin[a,b,c]

aState Key Laboratory of Agricultural Microbiology, Huazhong Agricultural University, Wuhan, China
bCollege of Animal Medicine, Huazhong Agricultural University, Wuhan, China
cHubei Provincial Key Laboratory of Preventive Veterinary Medicine, Huazhong Agricultural University, Wuhan, China
dCollege of Informatics, Huazhong Agricultural University, Wuhan, China
eSection of Paediatric Infectious Disease, Imperial College London, St Mary's Campus, London, United Kingdom

**ABSTRACT** Identification of microbial functional association networks allows interpretation of biological phenomena and a greater understanding of the molecular basis of pathogenicity and also underpins the formulation of control measures. Here, we describe PPNet, a tool that uses genome information and analysis of phylogenetic profiles with binary similarity and distance measures to derive large-scale bacterial gene association networks of a single species. As an exemplar, we have derived a functional association network in the pig pathogen *Streptococcus suis* using 81 binary similarity and dissimilarity measures which demonstrates excellent performance based on the area under the receiver operating characteristic (AUROC), the area under the precision-recall (AUPR), and a derived overall scoring method. Selected network associations were validated experimentally by using bacterial two-hybrid experiments. We conclude that PPNet, a publicly available (https://github.com/liyangjie/PPNet), can be used to construct microbial association networks from easily acquired genome-scale data.

**IMPORTANCE** This study developed PPNet, the first tool that can be used to infer large-scale bacterial functional association networks of a single species. PPNet includes a method for assigning the uniqueness of a bacterial strain using the average nucleotide identity and the average nucleotide coverage. PPNet collected 81 binary similarity and distance measures for phylogenetic profiling and then evaluated and divided them into four groups. PPNet can effectively capture gene networks that are functionally related to phenotype from publicly prokaryotic genomes, as well as provide valuable results for downstream analysis and experiment testing.

**KEYWORDS** functional association network inference, phylogenetic profiling, prokaryotic genome, *Streptococcus suis*, dereplication

The identification of functional association networks, i.e., correlative genes encoding protein complexes or involving common biological processes, allows novel virulence gene associations and mechanisms of pathogenicity to be elucidated (1, 2). In addition, networks of functional association can be used to predict the function(s) of uncharacterized proteins (3). Although genomewide surveys of functional links remain experimentally challenging in many organisms, e.g., protein complex purification, double mutant phenotyping, and correlated gene expression, etc. (4), advances in modern experimental technologies using high-throughput biology, such as next-generation sequencing and microarrays, have made it possible to capture the complex interplay between molecules.

Gene coexpression networks (GCNs), namely, transcript-transcript association networks, are typically generated by high-throughput methods for differential coexpression analysis of gene expression data generated, for example, by microarray or transcriptome

Address correspondence to Hui Jin, jinhui@mail.hzau.edu.cn.
The authors declare no conflict of interest.

sequencing, and are usually represented as an undirected graph (5, 6). GCNs in bacteria are typically constructed from transcriptome data, whereby gene sets or modules that exhibit a similar expression behavior across various environmental conditions, such as the invasion of host cells and tissues, heat shock, anaerobic stress, or iron restriction (5, 7–9). However, there are some limitations in terms of expression-based network inference in bacteria. For example, GCNs are typically established under specific experimental conditions, and not all transcriptional regulatory networks will be functional. In addition, because of the high cost of library construction and sequencing, publicly available transcriptomic data for some bacterial species is limited or not available, especially for bacterial field isolates (10).

While transcriptome data for some bacterial species is nonexistent or of limited availability, there are >360,000 sequenced bacterial genomes currently accessible to date (https://www.ncbi.nlm.nih.gov/genome/browse/#!/overview/), and these provide a convenient and cost-effective resource for constructing association networks based on the phylogenetic profiling method (3). Comparing the phylogenetic distributions is an effective way to predict the functional associations between nonhomologous genes, an approach first introduced by Pellegrini et al. (3). Typically, this method uses phylogenetic profiling between different species (11–13) and is rarely used with the same species. An important reason is that the evolutionary distance between the same species is too close and prevents identification of functional associations being identified via core genes phylogenetic profile analysis. Similarly, if there are too many isolates of the same strain, this hampers the construction of functional association networks. Another reason is that many species lack sufficient genomic data for comparison. However, recent advances, such as the recognition of considerable intravariation in phenotypes within a single species, e.g., physiological-biochemical characteristics, pathogenicity, and antibiotic resistance, and the current and rapidly increasing availability of whole-genome sequence data, create an environment facilitating the identification of functional association networks through phylogenetic profiling.

Here, we present PPNet (https://github.com/liyangjie/PPNet), the first tool for deriving large-scale bacterial association networks of one species based on phylogenetic profiling. To demonstrate the utility of our approach, we used it to identify a virulence-related gene network in the zoonotic bacterial pathogen *Streptococcus suis*. PPNet demonstrated excellence performance based on the evaluation measures used—the AUROC (area under the receiver operating characteristic), the AUPR (area under the precision-recall), and the overall score—and specific networks were validated by bacterial two-hybrid analysis, demonstrating its utility. The results suggest that PPNet offers a general approach to constructing microbial association networks by drawing upon easily acquired genome-scale data.

## RESULTS

**PPNet overview.** PPNet is implemented as a Python script and can be easily installed on Linux, MacOS, and Windows Subsystem for Linux (WSL) platforms. An overview of the PPNet workflow is shown in Fig. 1; further details are provided in Materials and Methods below. Briefly, PPNet requires both genome sequence and knowledge of the phenotype (e.g., pathogenic or nonpathogenic) of all strains as the input data. The first step is to perform quality control for all genomes to reduce data redundancy and biased genomes. Next, genome annotation of the high-quality genome data set is performed, and predicted protein sequences for each genome are extracted for gene clustering. Then, for all orthologs, a preliminary phylogenetic profile is generated across all isolates. In addition, PPNet divides strains into two groups based on phenotypic information provided by the user and compares the distribution of each ortholog from different phenotypic groups; only the phylogenetic profile of orthologs with significantly different distributions is selected for network inference (Fig. 2a). Finally, PPNet calculates the association coefficients among the genes based on the similarity of their phylogenetic profiles. By default, PPNet will list the association coefficient between each pair of genes that is less

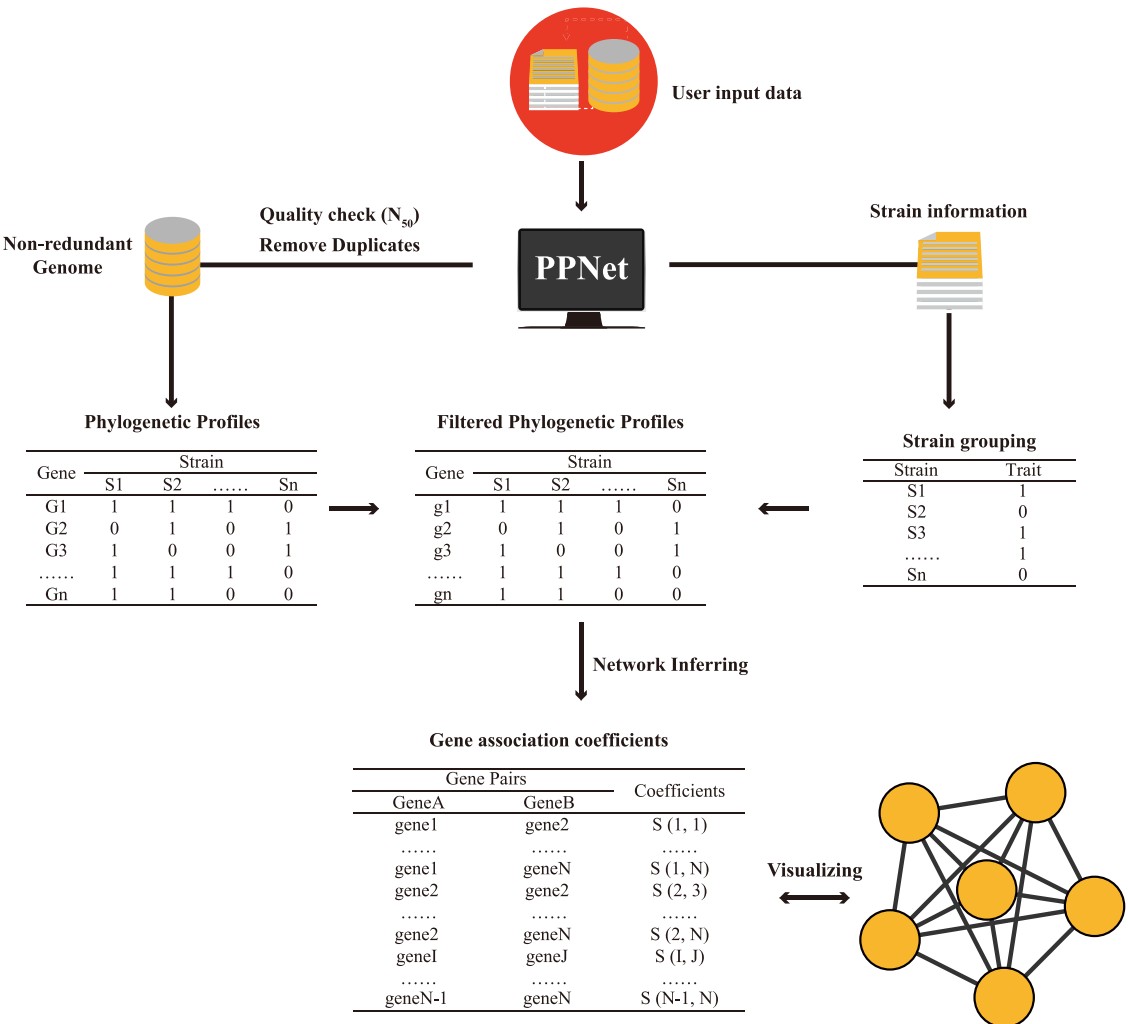

**FIG 1** Schematic representation of the PPNet workflow. Taking genome data and grouping information of strains as the input, each genome data goes through a set of filtration steps, including the removal of poor-quality genomes based on $N_{50}$ and the removal of duplicate genomes based on the ANI and ANC, with the thresholds for each step set by the user. Next, the obtained high-quality genomes are automatically annotated, and a preliminary phylogenetic profile is constructed. The phylogenetic profile is represented by a binary matrix, where each row represents an ortholog, each column represents a strain, and the "1" or "0" in each row refers to the presence or absence of the ortholog in each strain, respectively. The preliminary phylogenetic profile is then filtered by using the Fisher exact test; only the phylogenetic profile of orthologs with significant differences in distribution across strain groups is retained. Finally, the association coefficients among the orthologs are calculated based on the similarity of their phylogenetic profiles. These results are saved as the output and can be imported to Cytoscape for visualization.

than or equal to the first percentile, and the output list can be further visualized by Cytoscape (14).

**_S. suis_ virulence-related gene association network.** To demonstrate its usefulness, PPNet was used to infer the virulence-related gene association network of _S. suis_ from publicly available data. A total of 1,288 published _S. suis_ genome sequences, including 43 complete and 1245 draft genomes, were obtained from the National Center for Biotechnology Information (NCBI) FTP server (see Table S1 in the supplemental material). Based on the average nucleotide identity (ANI) and serotype, 34 genomes were identified as not being derived from _S. suis_ and were removed from further analyses. Thus, 1,254 _S. suis_ genomes were used as input data for PPNet.

After quality control and removal of redundant genome data by PPNet, 551 non-redundant and high-quality genomes were finally used for subsequent analysis. The preliminary phylogenetic profile created by PPNet contained 15,722 orthologs, with

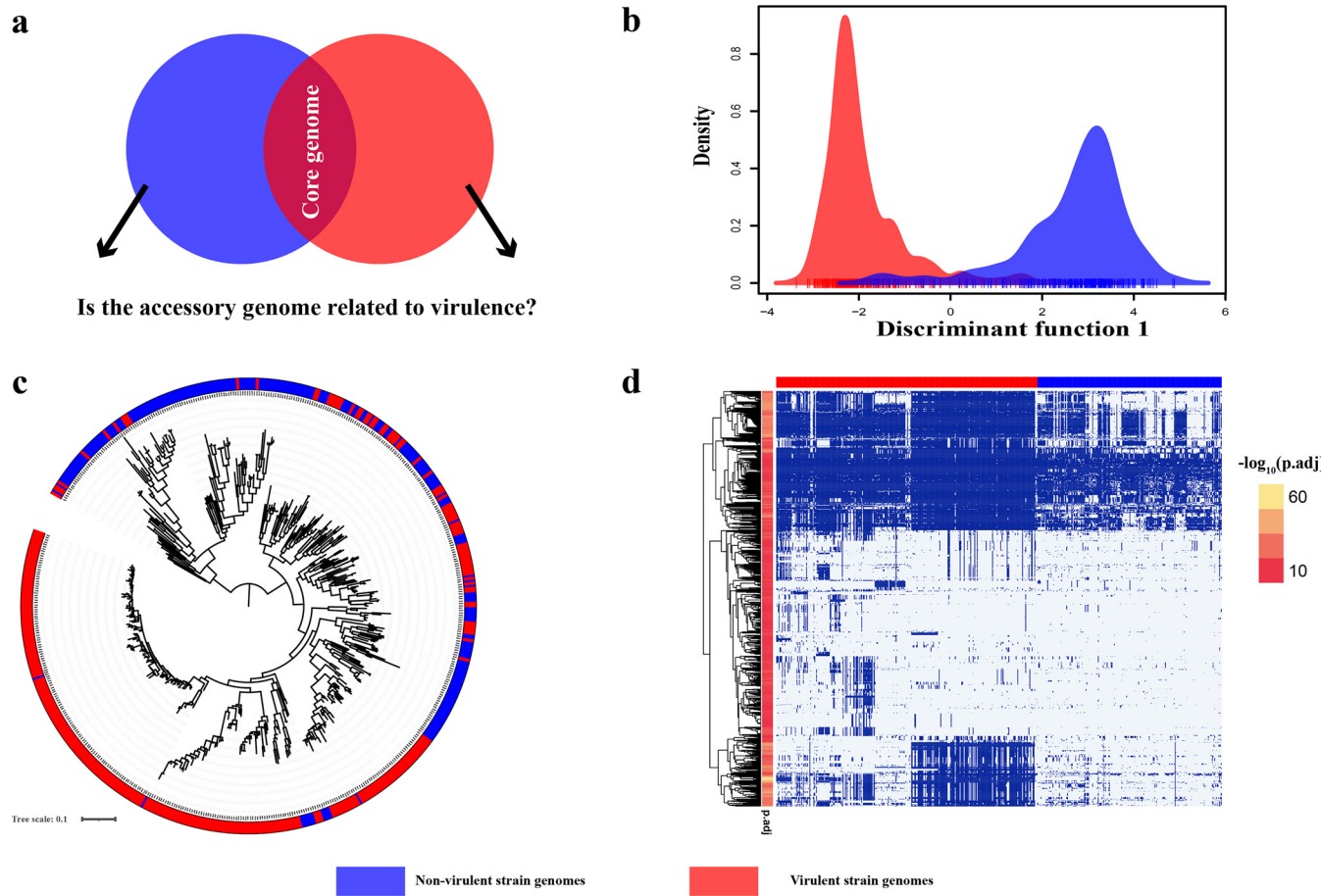

**FIG 2** Differences between virulent and nonvirulent serotype. (a) Venn diagram showing the premise for using the virulent serotype collection (red) and nonvirulent serotype collection (blue) of *S. suis* genomes. (b) DAPC was used to evaluate the separation between the isolates of the virulent serotype collection (red) and the nonvirulent serotype collection (blue) of *S. suis*, using the presence/absence data for genes in the accessory genome. (c) Phylogenetic tree of 551 *S. suis* strains based on the binary presence or absence of accessory genes. The outer ring provides information about the virulent (red) and nonvirulent serotypes (blue), respectively. (d) Heatmap visualizing the distribution of VRDGs in the genomes of *S. suis*. The presence (dark blue areas) or absence (light blue areas) of 1,060 VRDGS is shown in the heatmap. Each row indicates a VRDG and was clustered by hierarchical clustering based on the VRDG distribution. Each column indicates a strain, which was divided into virulent group (red) and nonvirulent group (blue). A colored strip from red to yellow shown on left side of the heatmap correspond to $-\log_{10} (P_{adj})$ values from low to high. The $P_{adj}$ values are the adjusted $P$ values computed by Fisher exact test under the null hypothesis that the presence or absence of this gene is unrelated to virulence and adjusted by false discovery rate.

1,141 and 14,581 genes assigned to the core (present in 99% of isolates) and accessory (variably present) genomes, respectively.

Capsular serotype and virulence of *S. suis* are known to be related (15). Accordingly, 323 *S. suis* strains of serotypes 1 to 5, serotypes 7 to 9, serotype 1/2, serotype 14, serotype 16, serotype 24, and serotype Chz were categorized here as the virulent group, while 228 strains of other serotypes, including serotype 6, serotypes 10 to 13, serotype 15, serotypes 17 to 19, serotype 21, serotype 23, serotype 25, serotypes 27 to 31, novel capsular polysaccharide loci (NCL), and nontypeable strains, were classified as the nonvirulent group (16–30). In addition, to determine the genetic diversity present in the accessory genomes associated with virulent and nonvirulent serotypes, we performed discriminant analysis of principle components (DAPC), which showed a clearcut separation between virulent and nonvirulent serotypes in terms of accessory genomes (Fig. 2b), suggesting that molecular serotyping was feasible for classification of virulent strains. Further, a phylogenetic tree (Fig. 2c) was constructed according to the binary presence or absence of accessory genes (31). The red and blue columns in the figure represent virulent and nonvirulent serotype strains, respectively. A group separation was observed between virulent and nonvirulent serotypes with a few exceptions, suggesting that the virulent phenotype classification through molecular serotyping was associated with accessory genome.

To obtain more valuable phylogenetic profiles, PPNet identifies virulence-related differential genes (VRDGs) by comparing the distribution of genes from virulent and nonvirulent genomes (Fig. 2a). Each gene receives its own null hypothesis of no association to virulence, and a Fisher test is performed (see Materials and Methods). VRDGs are defined as gene families that are overrepresented in virulent genomes. A total of 1,060 VRDGs were identified, and phylogenetic profiles of VRDGs were used to infer an association network (see Table S2). Figure 2d shows that VRDGs were predominantly present in virulent genomes compared to nonvirulent genomes. Finally, PPNet generated a total of 81 virulence-related gene association networks based on 81 binary similarity and dissimilarity measures (32, 33) (see Table S3).

**Performance of network association inference methods.** To evaluate these 81 networks of *S. suis*, the gene interaction networks of *S. suis* 05ZYH33 in the STRING (v11.0) was set as the gold standards for performance evaluation (34). We assessed the performance of the methods used for *S. suis* based on the AUROC, the AUPR (35), and the overall score, all of which have been used to summarize the performance of networks (36) (Fig. 3a; see also Table S3). The overall score and the performance of each network for all applied 81 binary similarity and dissimilarity measures are shown in Fig. 3a. Classification into the same cluster was made when the same AUROC, AUPR, and overall score were obtained from the different equations. Equation 60 (see Table S3) [$S_{OCHIAI-II} = ad/((a + b)(a + c)(b + c)(c + d))^{0.5}$], was found to give the highest value of overall score (see Fig. 3a).

In order to assess the capability of each equation, a scatterplot of minimum distances of the ROC (receiver operating characteristic) curve to the theoretical optimum point and AUROCs corresponding to the 81 equations for constructing the gene interaction networks of *S. suis* was generated (Fig. 3b). Based on the scatterplot, the final 43 dots obtained from 81 equations were divided into four groups (G1, G2, G3, and G4) according to the result of hierarchical clustering (see Fig. S1). The well-performing equations (see Fig. 3a) with least minimum distances and the highest AUROC scores were obtained in G1, which consisted of equations 60, 51, 52, 53, 54, etc. (see Fig. 3b). The ROC and PR (precision-recall) curves generated using OCHIAI-II similarity are displayed in Fig. 3c and d.

**Functional enrichment of *S. suis* virulence-related gene association networks.** To gain insights into *S. suis* virulence-related genes interactions, the association network was built at a cutoff of 3,215 edges with 753 genes by OCHIAI-II similarity, responding to an estimated precision of 50% based on the gold standard of all predicted and experimentally validated interactions from the STRING database (34, 36) (see Fig. S2). We found that the *S. suis* virulence gene-related network has a modular structure; to determine whether there was a functional association between genes within these modules, we analyzed the identified modules in *S. suis* SC19 strain for enrichment of Gene Ontology terms. Of 52 network modules, 21 were highly enriched in molecular function (Fig. 4). For example, 17 *S. suis* virulence-related genes were highly enriched in multiple molecular functions, which included protein-N (PI)/phosphohistidine-sugar phosphotransferase activity (B9H01_05885; B9H01_05880 and B9H01_05890), kinase activity (B9H01_05910; B9H01_05890 and B9H01_05900), lyase activity (B9H01_05850; B9H01_05870), and D-glucosamine phosphotransferase system (PTS) permease activity (B9H01_05885; B9H01_05880). The function of the other genes of this module included gluconate 5-dehydrogenase, M13 family metallopeptidase, muramidase-released protein, preprotein translocase subunit YajC, RpiB/LacA/LacB family sugar-phosphate isomerase, bifunctional 4-hydroxy-2-oxoglutarate aldolase/2-dehydro-3-deoxy-phosphogluconate aldolase, LacI family DNA-binding transcriptional regulator, DUF5590 domain-containing protein, and a putative protein of unknown function. The data indicate that PTS systems are closely involved in virulence with other enriched modules, e.g., kinase, lyase, and D-glucosamine PTS permease activities. The inferred associated networks also provided a list of functional predictions for *S. suis* uncharacterized genes for analyzing complex regulatory networks for further study.

**Experimental support for selected identified network associations.** To validate the network predicted from PPNet, we experimentally tested all of the two- and three-gene interaction modules identified in the virulence-related gene association networks

                                                  

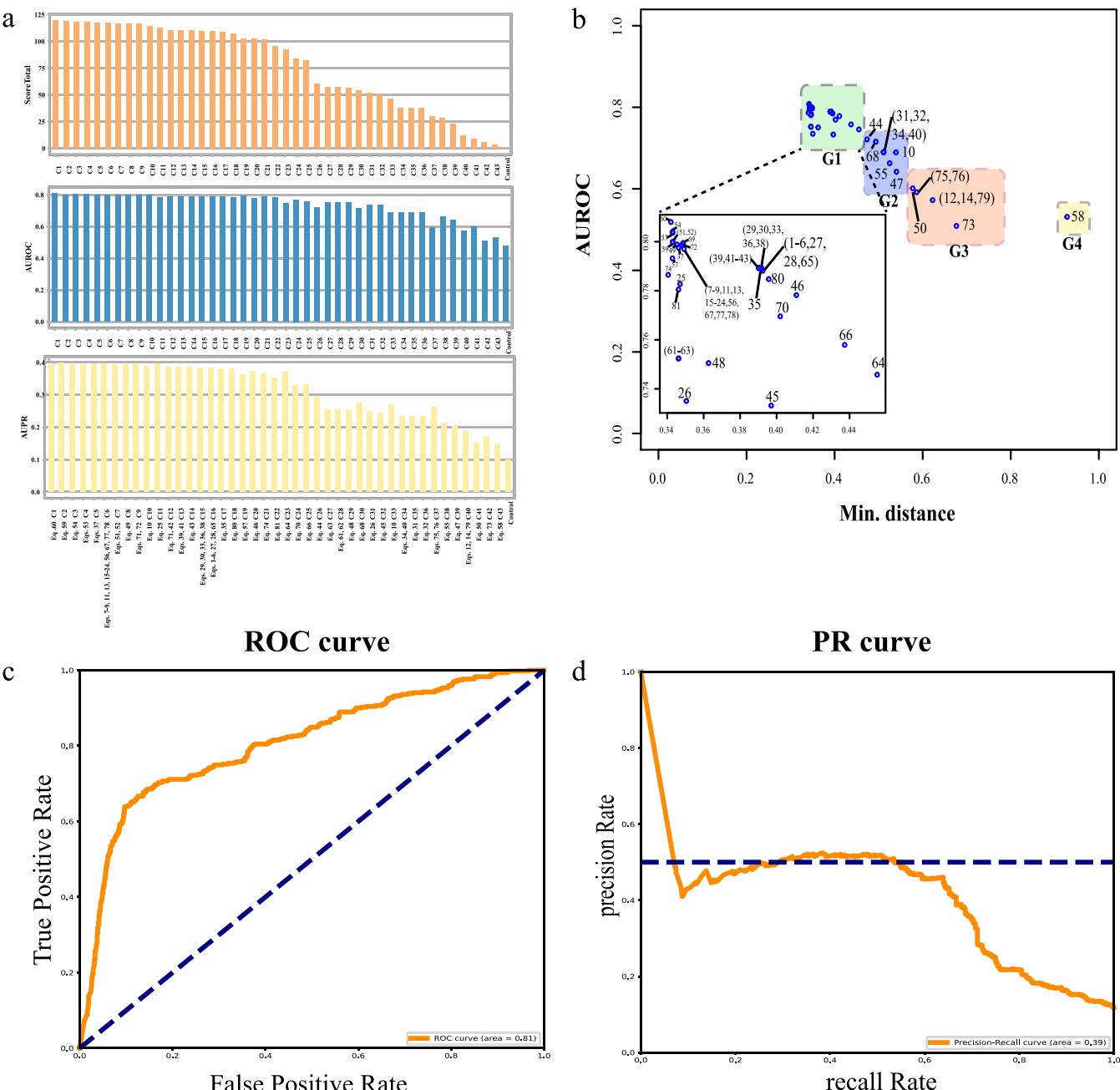

**FIG 3** Performance of network inference methods. (a) Assessment of network inference methods listed in Table S$3 in the supplemental material. Performance for the association networks of *S. suis* constructed by different binary similarity coefficients are indicated by the area receiver operating characteristic (AUROC) (blue), the area under the precision-recall (AUPR) (yellow), and the overall score (orange). The cluster included all the binary similarity coefficients sharing the same value of AUROC, AUPR, and the overall score. (b) A scatterplot depicts the minimum distance versus AUROC by 81 binary similarity and distance measures. According to the distribution, the final 43 dots obtained from 81 equations were divided into four groups (G1, G2, G3, and G4) through hierarchical clustering (see Fig. S1), represented by four different colored boxes. OCHIAI-II similarity (equation 60 [see panel a]) present in the first group had a relatively short minimum distance and the second-highest AUROC value. (c and d) AUROC (c) and PR (d) curves, as determined by OCHIAI-II similarity (equation 60 [see panel a]).

constructed by OCHIAI-II similarity, as predicted for *S. suis* SC19, using bacterial two-hybrid analysis. We selected 17 pairs of predicted two-gene interactions and six groups of predicted three-gene interactions and tested each of them individually by bacterial two-hybrid analyses (see Fig. S3). Thirty-five pairs of interactions among 52 genes were verified (Fig. 5; see also Fig. S3).

Predicted interactions were considered confirmed if they showed red clones on MacConkey-maltose indicator plates (37). A well-defined difference between positive

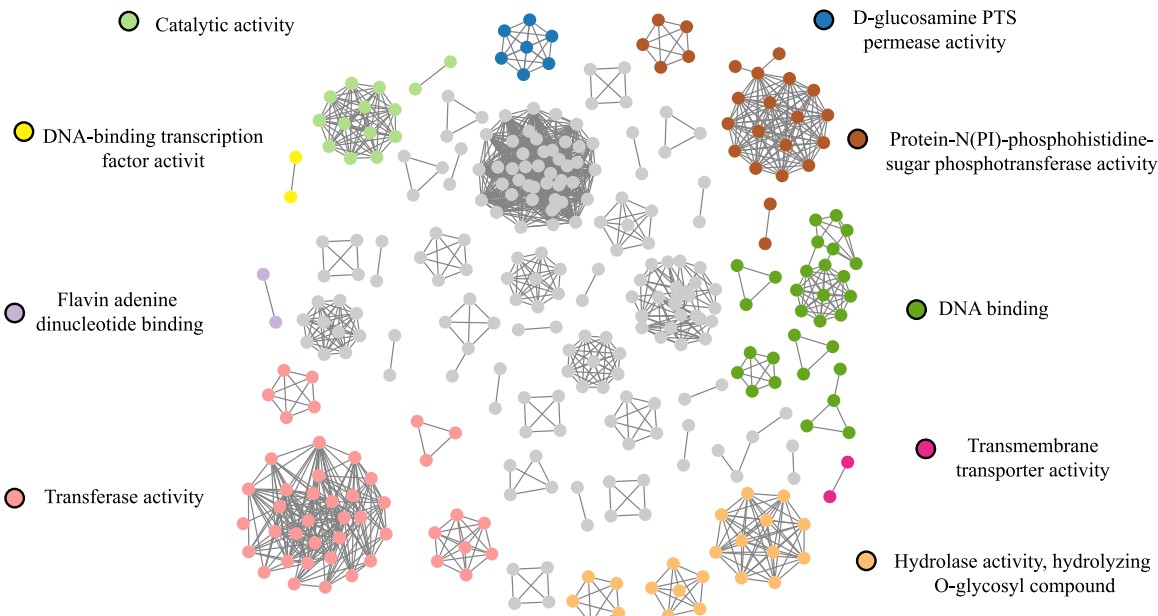

**FIG 4** VRDGs association network of *S. suis* SC19. The VRDGs association network of SC19 connects 1,508 interactions with 329 genes for *S. suis*, which was extracted from the whole network of Fig. S1. Gene Ontology term enrichment was performed for the network modules, and gray genes are those with no enrichment.

and negative results was displayed for all of the predicted interactions. Of 35 pairs, 21 (60%) showed positive results (Fig. 5). A total of two novel targets among 10 estimated interactions, not predicted or identified in the STRING database, displayed positive results identified by bacterial two-hybrid analyses. Using $\beta$-galactosidase assays, we also quantified the extent of protein interactions (37). Consistent with the results of bacterial two-hybrid analyses, all positive interactions showed high $\beta$-galactosidase activity indicative of interaction (Fig. 5).

Overall, the results indicate that PPNet can be used to predict novel virulence gene association networks, is complementary to those predicted in the STRING database, and provides an important theoretical starting point for studying the pathogenic mechanisms or other biological pathways of pathogenic bacteria.

## DISCUSSION

Here, we describe PPNet for the prediction of functional associations between nonhomologous genes, which can effectively capture gene networks that are functionally related to phenotype (e.g., pathogenic, antibiotic resistance, thermophilic, etc.), including operons, protein complexes, transcription factors and their target genes, etc. Several features distinguish PPNet from previous approaches (11–13). First, it can utilize genome data from a single species as the input. Fewer genes were identified with lower levels of similarity among genes in multiple microbial organisms by phylogenetic profiling (38) compared to in one bacterial species. More abundant homologous genes with high similarity among multiple strains of one species found by phylogenetic analysis should allow the construction of meaningful association networks. Second, the functional associations network identified by PPNet are closely linked to the phenotype, allowing a better understanding of the mechanisms that underly phenotypic differences. Third, a total of 81 binary similarity and distance measures were packaged in PPNet for users to choose from, since the choice of an appropriate similarity or distance measures is necessary for dealing with multivariate data represented by binary feature vectors (39). In order to demonstrate the utility of our approach, we used PPNet to determine whether virulence-related gene association networks could be identified from the publicly available genomes of the zoonotic pathogen *S. suis*.

Many bacteria have fast mutation rates, which are common in nature or hosts. Here, we chose *S. suis* as an example because of the availability of many genomes and the

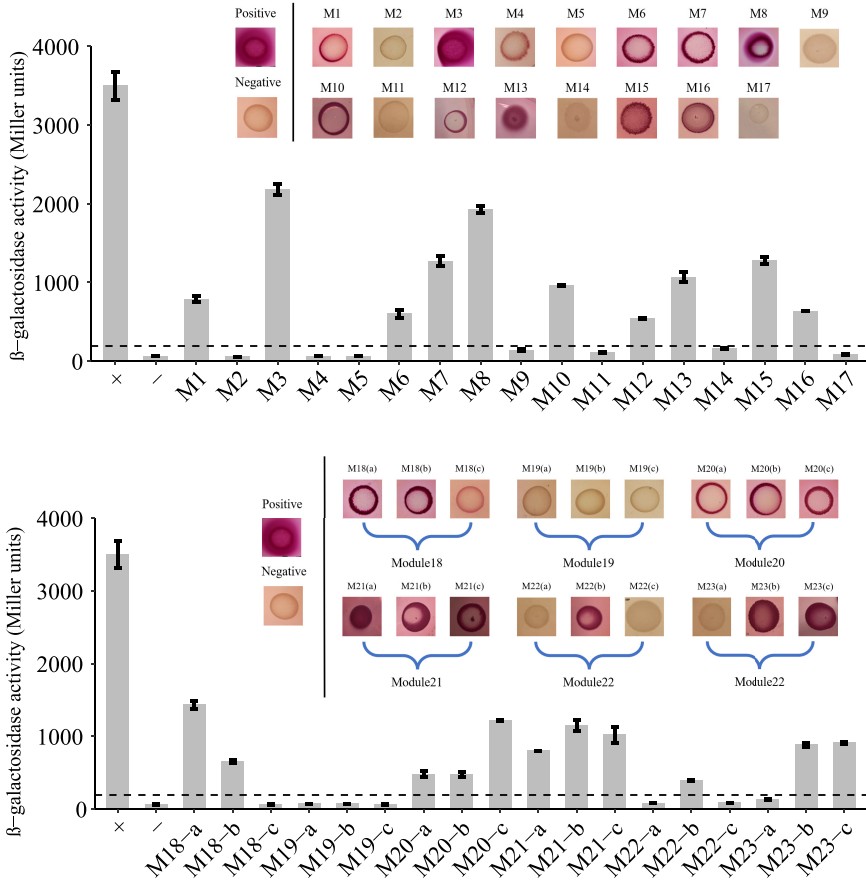

**FIG 5** Evaluation VRDGs association network by bacterial two-hybrid. Identification of the interactions of *S. suis* SC19 between the genes of two-gene interaction modules and the three-gene interaction modules fused to pKT25 and pUT18 by bacterial two-hybrid analyses was assessed in MacConkey-maltose indicator plates assay and by *β*-galactosidase activity assays in *E. coli* BTH101. BTH101 with pKT25 and pUT18 was the negative control. The inset displays the results of the MacConkey-maltose indicator plates assay. The "Mx" in the figure represents module x in Fig. S2 in the supplemental material. Positive colonies are red and negative colonies colorless. The bottom broken line indicates the cutoff value (ODc = 238.1481) for determining a positive of *β*-galactosidase activity, defined as three times the negative-control value. Error bars indicate the standard deviations (*n* = 3 biological replicates).

pathogenicity of different isolates can be variable. Moreover, as a nonmodel organism, more than half of the genes of its pangenome are uncharacterized and its pathogenic mechanisms are not fully understood. In a previous study, *S. suis* was divided into three groups (nonclinical, systemic, and respiratory) based on clinical data to investigate its genetic basis of disease (40). However, clinical information is missing for many genomic data and, on the other hand, *S. suis* as an opportunistic pathogen cannot be identified as nonpathogenic or pathogenic even if it is isolated from clinical health or disease cases. Previous studies show that different serotypes of *S. suis* have different pathogenic potential; strains isolated from diseased pigs mainly belong to certain serotypes (16–30). Hence, we determined here the serotypes of strains molecularly and then assigned them as virulent or nonvirulent based on the established relationship between serotype and virulence potential. DAPC showed a quite clear genetic difference in the accessory genomes between virulent and nonvirulent serotypes (Fig. 2b), suggesting our approach was valid.

Another program for determining genes associated with phenotypes is Kover, a k-mer-based software using machine learning algorithms that allows users to find some k-mers (sequences of k length) that are associated with phenotype (41). Kover recognizes k-mer presence/absence rather than gene presence/absence and is convenient for testing other types of representations for genomic variants, such as single

nucleotide polymorphisms (SNPs) and unitig level. However, Kover is less user friendly since users need to further annotate through sequence alignment to identify the cognate gene. Our approach and Scoary (42) both use the Fisher exact test to compare the significance of the presence/absence of genes associated with the different phenotypes, although these approaches do not detect SNPs in the orthologs of accessory genes because they are classified according to sequence similarity. In addition, Scoary considers the effect of population structure on gene distribution. However, Scoary is considered stringent, resulting in too few predictions (43), and is not conducive to identifying subsequent network inference studies based on phylogenetic profiles.

A total of 35 of 1,060 VRDGs identified in this study coincided with those of the 71 confirmed and putative *S. suis* virulence factors summarized previously (44). Of these, 24 VRDGs were involved in the virulence-related gene association networks on the basis of the genome of the SC19 strain of *S. suis* (Fig. 4). For example, *treR* have been shown *in vitro* to be related to virulence characteristics (45–47). It was also found that 32 genes were involved in interactions with *treR* in our association network (see Fig. S4). Three gene modules, including the ABC-type multidrug transporter gene *ccmA* (48) and the ABC-type amino acid transporter gene *hisM* (45) modules, were identified as positive by bacterial two-hybrid experiments in this study (see Fig. S4). In addition, the proposed virulence gene *scrR* was reported to be a repressor protein that is part of the sucrose operon (45). This gene was also found in our association networks, with three VRDGs being connected with it, suggesting that the VRDGs may be part of or allied to the sucrose operon (see Fig. S4). We also found another four genes associated with the *srtF* pilus gene (49–51) (see Fig. S4), suggesting that these genes may be related to flagellum synthesis.

PPNet also sought to determine whether any strains were replicates. The genomic data of *S. suis* strains downloaded from NCBI FTP server potentially contains replicate strains because of the isolation methods used. For example, in one study, six colonies were selected per swab of the same pig in China, and three were selected in United Kingdom. Isolates with alpha-hemolytic activity and positive biochemically results were stored as *S. suis* and then sent for whole-genome sequencing (52), leading to a possibility that multiple colonies isolated from the same pig are the same strain. Also, swabs from multiple pigs could be from the same pig farm. Although thousands of *S. suis* strains have had their whole genomes sequenced, there is region and serotype bias. For example, a total of 379 isolates from Vietnam were isolated and sequenced from 2015 and 2016 (see Table S1), suggesting the presence of multiple replicated genomes, which has the potential to confound statistical or probability analysis (53). Therefore, we reasoned that it was prudent to carry out a preliminary screening for potential replicated genomes in the NCBI database.

The rationale for our approach is that under evolutionary pressure, functionally related genes encode proteins that form a complex involved in carrying out reactions in the same biochemical pathway tend to cooccur or to coabsence in the genomes of different strains within a species. The outcome of network inference therefore varies from binary data and can be highly complementary to expression-based network inference. Binary similarity distance measures play an important role in the processing of binary data (32). Tremendous efforts have been made to find the most meaningful binary similarity and distance measures, which have been proposed in various fields, including biology, ethnology, taxonomy, image retrieval, geology, and chemistry (32, 54). Here, 81 binary similarity and distance measures, including 76 similarity and dissimilarity measures used over the last century showing a meaningful performance in their respective fields (54), and five new binary similarity coefficients (33), were used. In order to evaluate the different similarity and measures, STRING, a recognized gold standard, was applied to collect, score, and integrate all publicly available sources of protein-protein interaction information (34). Dependent on the ROC curve analysis (36, 55), PR curves (36, 56), overall score (35, 57), and clustering measures (32, 58), we found that OCHIAI-II similarity (equation 60) was the best for determining potential association networks of *S. suis* (see Fig. 3a).

In conclusion, our study had developed a powerful tool PPNet with 81 binary similarity and dissimilarity measures for network inference and evaluated PPNet from several aspects by constructing a functional association network for *S. suis*, which exhibits excellent performance based on evaluation measures, including AUROC, AUPR, and overall score, and selected examples were validated by bacterial two-hybrid experimental analysis.

One potential disadvantage of our binary similarity and distance measure approach is that the gene association networks identified do not include core genes. However, it should be noted that the accessory genome, believed to be important in phenotypic variation and genome evolution (59), is logically much bigger than the core genome, as in this study, especially as more genomes are added (Fig. 2b) (60), suggesting that this potential disadvantage is mitigated. Further work remains to be done to choosing a more appropriate gene phylogenetic profile to build the functional association network and to determine the threshold for judging whether two genes are related.

## MATERIALS AND METHODS

**Genome selection.** To obtain a more reliable genomic data, PPNet first calculates the $N_{50}$ values for all genomes; those with an $N_{50}$ of <10,000 are considered as comprising poorly sequenced or assembled data and are excluded from subsequent analysis. In addition, PPNet distinguishes genomes from the same strain according to the average nucleotide identity (ANI) and average nucleotide coverage (ANC) for each pair of genomes, MUMmer is used to align the input sequences, and the ANI and ANC are calculated by PYANI (61, 62). PPNet needs to set a threshold for ANI and ANC, respectively; genomes with ANI > threshold (ANI) and |1 − ANC| < threshold (ANC) values will be considered redundant genomes. Then, among the replicated genomes, we chose genomes of the isolates with a maximum $N_{50}$ as representative strains for further study. By default, PPNet will first test the numbers of nonredundant genomes identified by ANI at different thresholds and then select the inflection point as the threshold for ANI. The ANC threshold is then determined in the same way.

**Construction and filtering of the phylogenetic profile.** PPNet uses Prokka (63) to automate the annotation all genomes and then extracts the GFF3 format annotation files from the output files as the input files for Roary (31). To construct the phylogenetic profile, we adopted the default setting of Roary, which splits paralogs from homologous groups into groups of true orthologs by using conserved gene neighborhood information.

In order to obtain a more valuable phylogenetic profile for network prediction, PPNet requires strain grouping information and creates a 2 × 2 contingency table, the levels being presence and absence for the trait and gene, respectively, with counts of the numbers of isolates in each cell. For each gene, we assume the null hypothesis, i.e., it is independent of virulence and uses the Fisher exact test to compute *P* values. Finally, *P* values were corrected by using the false discovery rate. Genes with an adjusted *P* value ($P_{adj}$) of <0.05 are considered phenotype-related differential genes, and the phylogenetic profile of these differential genes was used for network inference.

**Network inference.** To investigate the relationship between phenotype-related differential genes, PPNet constructs association networks by conversion of the distribution of these genes by binary classification. Specifically, four variables—*a*, *b*, *c*, and *d*—are defined as follows: *a* is the number of genes present in group 1 and group 2; *b* and *c* are the numbers of genes present in group 1 but not present in group 2 and vice versa, respectively; and *d* is the number of genes where the gene is absent in both group 1 and group 2. Subsequently, 81 binary similarity and distance measures are used to construct phenotype-related differential gene networks. A detailed description of all binary similarity and distance coefficients is given in Table S3.

**Genome sequences of *S. suis*.** Draft or complete genome sequences of 1,288 *S. suis* were downloaded from the NCBI FTP server (April 2019; ftp://ftp.ncbi.nlm.nih.gov/genomes/). To facilitate subsequent analysis, the 1,288 *S. suis* strains were renamed as SS001 to SS1288. (Details of the 1,288 *S. suis* strains can be found in Table S1.) To determine whether all of the genomic data collected belonged to *S. suis*, we used the Python module PYANI (62) to calculate the ANI among 1,288 *S. suis* genomes (64). The ANI was obtained by using MUMmer (NUCmer) to align the input sequences (61). If the ANI between the strain and any serotype reference strain of *S. suis* (see Table S4) was >95%, the strain was designated to be a member of *S. suis* (64).

**Molecular serotyping of *S. suis*.** We used genome sequences of all *S. suis* as the query in BLASTn searches against a nucleotide BLAST database of all serotype-specific genes in the CPS synthesis locus (17, 65–68) for molecular serotyping. The serotype-specific gene analysis identified 33 classic serotypes, except for two pairs of serotypes: (i) serotypes 1 and 14 and (ii) serotypes 2 and 1/2, which have no antigenic differences genes between them (17) (see Table S5).

**Grouping of *S. suis* into virulent and nonvirulent groups.** To find the virulence-related genes of *S. suis*, we divided the *S. suis* strains into virulent and nonvirulent groups according to epidemiological surveys based on their serotypes (16–30). Specifically, strains of serotypes 1 to 5, serotypes 7 to 9, serotype 1/2, serotype 14, serotype 16, and serotypes 24 and Chz were considered highly virulent, while the remaining strains of other serotypes, including serotypes 10 to 13, serotype 15, serotypes 17 to 19, serotype 21, serotype 23, serotype 25, and serotypes 27 to 31, were classified as nonvirulent (16–30). In addition, NCL and nontypeable strains were also classified as nonvirulent since they are mainly isolated from healthy carrier pigs (16).

**DAPC.** Discriminant analysis of principle components (DAPC) was implemented in the R package adegenet (69, 70) to determine whether genetically related individuals were closely grouped. In this study, we used the presence or absence of accessory genes in 551 *S suis* genotyped isolates to determine the differences between the high and low virulence groups as classified by virulent/nonvirulent molecular serotypes, as described below. After identifying the optimal number of principal components (PCs) by cross-validation, we retained 100 PCs based on the preliminary data, which accounted for approximately 82.73% of the total genetic variability, and all discriminant functions were retained (40).

The phylogenetic tree was created using the *S. suis* accessory genome by FastTree to group isolates together based on the presence or absence of genes in their accessory genomes (71). This phylogenetic tree was visualized with iTOL and rerooted using the midpoint rooting method (72).

**Network performance.** In order to evaluate the performance of the binary similarity and distance measures, we used STRING (https://string-db.org) as the gold standard. A total of 81 similarity coefficients were assessed by using the area under the receiver operator curve (AUROC), the area under the precision versus recall curve (AUPR), and an overall score that summarizes the performance across the 81 networks (35, 36). The overall score is defined as the mean specific $P$ value (log transformed) of the network, which was used in the previous DREAM challenge (35, 57).

$$\text{overall score} = \frac{-\log_{10} P_{\text{ROC}} - \log_{10} P_{\text{PR}}}{2}$$

The negative-control group was generated randomly by setting the expected probability of interaction as 0.5 using the Python random package (36). The minimum distance of the ROC curve to the theoretical optimum point and AUROC were used to evaluate the performance of the groups of equations (32), and hierarchical clustering was performed by using the hclust function in R stat package.

To visualize the VRDG association networks of *S. suis* based on the binary similarity coefficient with the best overall score, we constructed high-confidence networks at an estimated precision of 50%. The network modules were annotated with Gene Ontology term by eggNOG-mapper (73, 74) and enriched by ClusterProfiler (75).

**Experimental materials and design.** A total 14 pairs of predicted two-gene interactions and eight groups of predicted three-gene interactions and a pair of positive controls were tested (see Fig. S3). Among these interactions, six pairs of novel interactions between two genes modules and ten pairs of novel interactions among three gene modules, which were not predicted in the STRING database, were selected for experimental validation.

Strains and plasmids used in this study are listed in Table S6 in the supplemental material. *S. suis* strains were grown in tryptic soy broth supplemented with 10% bovine serum at 37°C under vigorous agitation. *Escherichia coli* BTH101strain was grown aerobically in lysogeny broth at 37°C. Bacterial two-hybrid analyses, including a MacConkey-maltose indicator plate assay and $\beta$-galactosidase assays, were performed as described in the manual of the bacterial adenylate cyclase two-hybrid system kit (Euromedex).

Each interaction pair was scored on MacConkey-maltose indicator plate assays on a minimum of three individual occasions, and $\beta$-galactosidase assays were performed at least three times.

**Data availability.** Reannotated genome data in GFF format of 551 nonredundant *S. suis* strains for construction of the pangenome have been be deposited at Cyverse (https://de.cyverse.org/dl/d/8354EE88-08A6-46D5-9241-FC5BB91E349C/gff_file.zip).

## SUPPLEMENTAL MATERIAL

Supplemental material is available online only.
**SUPPLEMENTAL FILE 1**, PDF file, 4.7 MB.
**SUPPLEMENTAL FILE 2**, XLSX file, 2 MB.

## ACKNOWLEDGMENTS

We thank Qi Huang (College of Animal Medicine, Huazhong Agricultural University, Wuhan, China) for providing strains and plasmids used in the bacterial two-hybrid analyses.

This study was supported by grants from the National Key Research and Development Program of China (2021YFD1800400), the Natural Science Foundation of Hubei Province (2021CFA016), the Hubei Province Natural Science Foundation for Distinguished Young Scholars (2020CFA060), the Applied Basic Research Project of Wuhan (grant 2020020601012254), and the UK Biotechnology and Biological Sciences Research Council (BB/S019901/1).

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
