## [Reviewer comments · Microbiology Spectrum]

Microbiology Spectrum

PPNet: Identifying functional association networks by phylogenetic profiling of prokaryotic genomes

Yangjie Li, Bin Ma, Kexin Hua, Huimin Gong, Rongrong He, Rui Luo, Dingren Bi, Rui Zhou, Paul Langford, and Hui Jin

Corresponding Author(s): Hui Jin, Huazhong Agricultural University

Review Timeline:

Submission Date:	September 21, 2022
Editorial Decision:	October 13, 2022
Revision Received:	November 29, 2022
Accepted:	December 1, 2022

Editor: Sébastien Faucher

Reviewer(s): Disclosure of reviewer identity is with reference to reviewer comments included in decision letter(s). The following individuals involved in review of your submission have agreed to reveal their identity: Adonis D'Mello (Reviewer #2)

Transaction Report:

DOI: <https://doi.org/10.1128/spectrum.03871-22>

October 13, 2022

Dr. Hui Jin
Huazhong Agricultural University
State Key Laboratory of Agricultural Microbiology, College of Veterinary Medicine, Huazhong Agricultural University
Wuhan
China

Re: Spectrum03871-22 (PPNet: Identifying functional association networks by phylogenetic profiling of prokaryotic genomes)

Dear Dr. Hui Jin:

Your manuscript was reviewed by two expert in the field. They both find that this new tools has merits and should be useful. They raised a few questions and issues that needs to be addressed before your manuscript can be considered for publication. In addition to addressing the reviewers comment, please modify Figure S1 to make the labels readable.

Link Not Available

Sincerely,

Sébastien Faucher

Journals Department
Reviewer comments:

Reviewer #1 (Comments for the Author):

The manuscript describes a bioinformatics tool, PPNet, which makes it possible to make associations between certain phenotypes and genes in bacteria. Although the idea is interesting, I have a few comments:

1. As I understand the tool relies heavily on Roary to find homology between genes and build phylogenetic relationships. I find it a bit risky that the core of a new tool depends on a pipeline that hasn't been supported for 4 years (soon 5 years). In my opinion, this compromises the longevity of PPNet.

2. It is unclear how PPNet handles paralogs. What are the criteria for having only orthologous genes and excluding paralogs?
3. There is a tool, kover (<https://aldro61.github.io/kover/index.html>), which uses a machine learning approach and also allows genomic sequences to be associated with phenotypes. Authors should discuss the advantages of their tools over published ones.
4. PPNet requires that strain groups be created by the user (virulent vs non-virulent, for example). Sometimes it can be difficult to categorize certain strains. How resilient is PPNet? In this sense, if a strain is misclassified, what would be the impact on the results?
5. I believe Figure 2a is not cited in the text.
6. The authors have chosen to place non-typeable strains of *S. suis* in the non-virulent group. Why? Is it so impossible for virulent strains to be untypable?

Reviewer #2 (Comments for the Author):

Summary

Li et al. have developed PPNet, a useful bioinformatic tool, implemented in python, to classify pairs of functionally related bacterial proteins, within a bacterial species, using what is referred to as "phylogenetic profiles" (a binary matrix of gene presence/absence) of all non-core proteins in the pangenome. They selected high quality publicly available *S. suis* genomes and assigned them into two groups of virulent and non-virulent strains based on in silico serotyping. These were then filtered for core genes and parts of the accessory genome were shown to correlate with virulent and non-virulent groups. Parts of the accessory genome were then constructed into functionally associated networks of orthologs, and some were validated using StringDB and experimentally using bacterial two-hybrid systems.

PPNet can be a useful functional tool for pangenomics and assignment of functional networks of non-core proteins. It is particularly useful for understudied species and accessory genomes are typically less studied than the core genome. It could be used in conjunction with GWAS either upstream or downstream. PPNet as a "phenotype-to-gene presence/absence" association tool is not a novel concept and more robust tools exist for the same in the microbial GWAS field. However, PPNet as a fast and efficient functional association network tool for relevant accessory genes, is a novel and valuable resource for researchers like myself and others as an important step in analysis and validation of virulent genes in diverse bacterial omics datasets.

Major Comments

- 1) Lines 139-144: It is unlikely that serotypes and virulence correlate as well as the paper suggests to classify the genomes into virulent and non-virulent. The authors attempted to work around this problem with DAPC but it needs to be further explained how the accessory genome was filtered into 1060 VRDGs either in the results or methods. Alternative to the DAPC method, authors could validate their phenotype associated orthologs with a more robust method other than selection of 100 validated principal components for DAPC. Authors have used the tool Roary for ortholog determination, but gene presence/absence microbial GWAS can be conducted using Scoary (or others), another tool by the same group for statistically significant phenotype associated separation of ortholog data. Since this is an already established concept in the phenotype association field, these tools should be used or compared in relation to PPNet, and any advantages or disadvantages of both methods should be highlighted.
- 2) The paper doesn't discuss or address the possibility of an ortholog of accessory genes having minor differences in sequence which associate with one phenotype over another or affect functional association in one group over the other. It is worth discussing this, as other tools operate at the SNP and unitig level to identify such variants. PPNet would likely not detect these differentially present gene networks as they would not separate out at the ortholog level.
- 3) While PPNet is simple to download and install as the paper suggests. Once installed it seems to be missing a key python script (`average_nucleotide_identity.py`) rendering it non-functional. Authors should fix the github version and attempt installation and execution of PPNet on an independent system.
- 4) The data presented in Figure 3 for choice of similarity/distance measure (Eqn. 60) doesn't match the data referenced in the Supplementary Table 2. In the table, AUROC and AUPR values exceed 100% and this has not been explained nor do the values always correspond to the ones in the figure. The chosen equation (C1 Eqn60) for the network construction is the only one with an AUROC >100% listed in the table.
- 5) Input datasets of binary accessory genomes (used in DAPC and network construction) could be provided as a supplemental table for review. This could also be of benefit to the people who study *S. suis* as a database of *S. suis* accessory orthologs.

Minor Comments

Figure Legend 1

- ANI and ANC thresholds are described as user-adjustable but the current version of PPNet appears to be hard coded.
- The figure clearly depicts what is meant by the term "phylogenetic profile". However, this should be better explained in the text as just a binary matrix as "phylogenetic profile" is a bit vague and could be referring to other things like phylogenetic tree topology.

Figure Legend 2

-Panel C: It needs to be mentioned that the tree is constructed using the accessory ortholog profile in the legend as it is mentioned in the text. This could be misinterpreted as a whole genome tree.

-Panel D has $-\log_{10}(\text{padj.})$ mentioned but it's not clear what these values reflect.

Figure Legend 3

-Panel a cites Supplemental Table 5 when it should be Supplemental Table 2

-Panel b highlights the relevance of Eqn. 60 for *S suis* but the colored boxes are not explained.

Figure Legend 5

-Cite the Supplemental table listing each "Mx" number and associated genes in the legend.

Line 38 - typo: experimental -> experiments

Lines 44-47 - same as the abstract

Line 128 - typo: publically -> publicly

Line 156-157 - significantly distributed is a misnomer as no direct gene presence-absence statistics were present. Perhaps "overrepresented in virulent genomes"

Staff Comments:

Preparing Revision Guidelines

Please return the manuscript within 60 days; if you cannot complete the modification within this time period, please contact me. If you do not wish to modify the manuscript and prefer to submit it to another journal, please notify me of your decision immediately so that the manuscript may be formally withdrawn from consideration by Microbiology Spectrum.

Summary

Li et al. have developed PPNet, a useful bioinformatic tool, implemented in python, to classify pairs of functionally related bacterial proteins, within a bacterial species, using what is referred to as “phylogenetic profiles” (a binary matrix of gene presence/absence) of all non-core proteins in the pangenome. They selected high quality publicly available *S. suis* genomes and assigned them into two groups of virulent and non-virulent strains based on *in silico* serotyping. These were then filtered for core genes and parts of the accessory genome were shown to correlate with virulent and non-virulent groups. Parts of the accessory genome were then constructed into functionally associated networks of orthologs, and some were validated using StringDB and experimentally using bacterial two-hybrid systems.

PPNet can be a useful functional tool for pangenomics and assignment of functional networks of non-core proteins. It is particularly useful for understudied species and accessory genomes are typically less studied than the core genome. It could be used in conjunction with GWAS either upstream or downstream. PPNet as a “phenotype-to-gene presence/absence” association tool is not a novel concept and more robust tools exist for the same in the microbial GWAS field. However, PPNet as a fast and efficient functional association network tool for relevant accessory genes, is a novel and valuable resource for researchers like myself and others as an important step in analysis and validation of virulent genes in diverse bacterial omics datasets.

Major Comments

1) Lines 139-144: It is unlikely that serotypes and virulence correlate as well as the paper suggests to classify the genomes into virulent and non-virulent. The authors attempted to work around this problem with DAPC but it needs to be further explained how the accessory genome was filtered into 1060 VRDGs either in the results or methods.

Alternative to the DAPC method, authors could validate their phenotype associated orthologs with a more robust method other than selection of 100 validated principal components for DAPC. Authors have used the tool Roary for ortholog determination, but gene presence/absence microbial GWAS can be conducted using Scoary (or others), another tool by the same group for statistically significant phenotype associated separation of ortholog data. Since this is an already established concept in the phenotype association field, these tools should be used or compared in relation to PPNet, and any advantages or disadvantages of both methods should be highlighted.

2) The paper doesn't discuss or address the possibility of an ortholog of accessory genes having minor differences in sequence which associate with one phenotype over another or affect functional association in one group over the other. It is worth discussing this, as other tools operate at the SNP and unitig level to identify such variants. PPNet would likely not detect these differentially present gene networks as they would not separate out at the ortholog level.

3) While PPNet is simple to download and install as the paper suggests. Once installed it seems to be missing a key python script (`average_nucleotide_identity.py`) rendering it non-functional. Authors should fix the github version and attempt installation and execution of PPNet on an independent system.

4) The data presented in Figure 3 for choice of similarity/distance measure (Eqn. 60) doesn't match the data referenced in the Supplementary Table 2. In the table, AUROC and AUPR

values exceed 100% and this has not been explained nor do the values always correspond to the ones in the figure. The chosen equation (C1 Eqn60) for the network construction is the only one with an AUROC >100% listed in the table.

5) Input datasets of binary accessory genomes (used in DAPC and network construction) could be provided as a supplemental table for review. This could also be of benefit to the people who study *S. suis* as a database of *S suis* accessory orthologs.

Minor Comments

Figure Legend 1

- ANI and ANC thresholds are described as user-adjustable but the current version of PpNet appears to be hard coded.
- The figure clearly depicts what is meant by the term “phylogenetic profile”. However, this should be better explained in the text as just a binary matrix as “phylogenetic profile” is a bit vague and could be referring to other things like phylogenetic tree topology.

Figure Legend 2

- Panel C: It needs to be mentioned that the tree is constructed using the accessory ortholog profile in the legend as it is mentioned in the text. This could be misinterpreted as a whole genome tree.
- Panel D has $-\log_{10}(\text{padj.})$ mentioned but it's not clear what these values reflect.

Figure Legend 3

- Panel a cites Supplemental Table 5 when it should be Supplemental Table 2
- Panel b highlights the relevance of Eqn. 60 for *S suis* but the colored boxes are not explained.

Figure Legend 5

- Cite the Supplemental table listing each “Mx” number and associated genes in the legend.

Line 38 – typo: experimental -> experiments

Lines 44-47 – same as the abstract

Line 128 – typo: publically ->publicly

Line 156-157 – significantly distributed is a misnomer as no direct gene presence-absence statistics were present. Perhaps “overrepresented in virulent genomes”

Response to reviewer's comments:

Reviewer #1:

The manuscript describes a bioinformatics tool, PPNet, which makes it possible to make associations between certain phenotypes and genes in bacteria. Although the idea is interesting, I have a few comments:

1. As I understand the tool relies heavily on Roary to find homology between genes and build phylogenetic relationships. I find it a bit risky that the core of a new tool depends on a pipeline that hasn't been supported for 4 years (soon 5 years). In my opinion, this compromises the longevity of PPNet.

Response: We appreciate your professional suggestion. It's true that Roary hasn't had a major version update in a long time, but Roary is still one of the most used canonical pan genome pipelines. We found the software has been used and cited more than 630 times in 2022 (data sourced from Google Scholar), including in not only Microbial Spectrum but other highly-rated journals, for example:

1. Lewin GR, Stocke KS, Lamont RJ, Whiteley M. 2022. A quantitative framework reveals traditional laboratory growth is a highly accurate model of human oral infection. Proc Natl Acad Sci U S A 119:1–11.
2. Bai X, Ylinen E, Zhang J, Salmenlinna S, Halkilahti J, Saxen H, Narayanan A, Jahnukainen T, Matussek A. 2022. Comparative Genomics of Shiga Toxin-Producing Escherichia coli Strains Isolated from Pediatric Patients with and

without Hemolytic Uremic Syndrome from 2000 to 2016 in Finland. *Microbiol Spectr* 10: e0066022

3. Yang Y, Nguyen M, Khetrapal V, Sonnert ND, Martin AL, Chen H, Kriegel MA, Palm NW. 2022. Within-host evolution of a gut pathobiont facilitates liver translocation. *Nature* 607:563–570.

4. Vassallo CN, Doering CR, Littlehale ML, Teodoro GIC, Laub MT. 2022. A functional selection reveals previously undetected anti-phage defence systems in the *E. coli* pangenome. *Nat Microbiol* 7:1568-79.

2. It is unclear how PPNet handles paralogs. What are the criteria for having only orthologous genes and excluding paralogs?

Response: We are sorry that we didn't clarify this important issue in the manuscript and we thank you very much for your constructive comments.

Paralogs are identified as genes with near identical sequence may perform a different function or under differential regulation. For this reason, we adopted the default setting of Roary, which splits paralogs from homologous groups into groups of true orthologs by using conserved gene neighborhood information. We have added this information to the "MATERIALS AND METHODS" section in the revised manuscript (lines 330-334):

"PPNet uses Prokka (63) to automate the annotation all genomes, and then extracts the GFF3 format annotation files from the output files as the input files for Roary (31) to construct the phylogenetic profile, we adopted the default

setting of Roary, which splits paralogs from homologous groups into groups of true orthologs by using conserved gene neighborhood information.”

3. There is a tool, kover (<https://aldro61.github.io/kover/index.html>), which uses a machine learning approach and also allows genomic sequences to be associated with phenotypes. Authors should discuss the advantages of their tools over published ones.

Response: We thank the reviewer for the constructive comments. Kover is an excellent software, and we have added some discussion to the revised manuscript (lines 243-249):

“Another programme for determining genes associated with phenotypes is Kover, a k-mer-based software using machine learning algorithms which allows users to find some k-mers (sequences of k length) that are associated with phenotype (41). Kover recognises k-mer presence/absence rather than gene presence/absence, and is convenient to test other types of representations for genomic variants, such as single nucleotide polymorphisms (SNPs) and unitig level. However, Kover is less user friendly as users need to further annotate through sequence alignment to identify the cognate gene.”

4. PPNet requires that strain groups be created by the user (virulent vs non-virulent, for example). Sometimes it can be difficult to categorize certain strains. How resilient is PPNet? In this sense, if a strain is misclassified, what

would be the impact on the results?

Response: Thank you very much for your constructive comments. The correct grouping of strains is important, therefore, PPNet requires the users to have certain understanding of their strains. If some strains are difficult to group due to lack of certain theoretic support, we suggest that these strains can be filtered out to reduce the impact on the results. In our study, we grouped strains according to the epidemiological surveys. However, epidemiology can only ensure that most strains can be grouped correctly, and there will inevitably be cases in the literature that are misclassified. Nevertheless, PPNet finally identified 1060 VRDGs based on Fisher's exact test, many of which have been confirmed by previous studies. In addition, the functional association network of *S. suis* predicted by PPNet was shown to be reliable by AUROC, AUPR, overall scoring and bacterial two-hybrid experimental. Therefore, PPNet is a resilient software.

5. I believe Figure 2a is not cited in the text.

Response: We sincerely apologize for our mistake. Figure 2a is a schematic diagram describing how phenotype related genes are identified, and has been cited in the revised manuscript (lines 102 -105, lines 138-140):

Lines 102 -105: "In addition, PPNet divides strains into two groups based on phenotypic information provided by the user, and compares the distribution of

each ortholog from different phenotypic groups, and only the phylogenetic profile of orthologs with significantly different distributions is selected for network inference (Fig. 2a).”

Lines 138-140: “To obtain more valuable phylogenetic profiles, PPNet identifies virulence-related differential genes (VRDGs) by comparing the distribution of genes from virulent and non-virulent genomes (Fig. 2a).”

6. The authors have chosen to place non-typeable strains of *S. suis* in the non-virulent group. Why? Is it so impossible for virulent strains to be untypable?

Response: Thanks for your valuable comments. We placed non-typeable strains of *S. suis* in the non-virulent group because Non-typeable *S. suis* strains are mainly isolated from healthy carrier pigs as reported in the article by Segura *et al.*:

“Nontypable *S. suis* strains are also frequently isolated, mainly from healthy carrier pigs.”

Segura M, Fittipaldi N, Calzas C, Gottschalk M. 2017. Critical *Streptococcus suis* Virulence Factors: Are They All Really Critical? *Trends Microbiol.* Elsevier Ltd <https://doi.org/10.1016/j.tim.2017.02.005>.

In view of the comment, we have added this information to the “MATERIALS AND METHODS” section in the revised manuscript (lines 369-376):

“To find the virulence-related genes of *S. suis*, we divided the *S. suis* strains

into virulent and non-virulent groups according to epidemiological surveys based on their serotypes (16–30). Specifically, strains of serotypes 1-5, serotypes 7-9, serotype 1/2, serotype 14, serotype 16, serotype 24 and Chz were considered highly virulent, while remain strains of other serotypes including serotypes 10-13, serotype 15, serotypes 17-19, serotype 21, serotype 23, serotype 25, serotypes 27-31 were classified as non-virulent (16–30). In addition, NCL and non-typable strains were also classified as non-virulent as they are mainly isolated from healthy carrier pigs (16)''

Reviewer #2:

Major Comments:

1. Lines 139-144: It is unlikely that serotypes and virulence correlate as well as the paper suggests to classify the genomes into virulent and non-virulent.

Response: Previous studies have shown that shows that different serotypes of *S. suis* have different pathogenic potential. Serotype 2 is the most commonly associated with diseases in pigs and humans (Segura et al., 2017). Other serotypes (1, 3-5, 7-9, 1/2, 14, 16, 24) are also routinely isolated from clinical pig cases (Liu et al., 2013). Therefore, we grouped strains according to the epidemiological surveys. Many other bacterial species have also found a correlation between serotype and pathogenicity, for example: *Actinobacillus pleuropneumoniae*,

Escherichia coli, *Glaesserella parasuis*, *Porphyromonas gingivalis*, *Streptococcus pneumoniae*, etc. Therefore, PPNet will be of interest to and usable by researchers interested in human and veterinary pathogens of worldwide importance.

Segura M, Fittipaldi N, Calzas C, Gottschalk M. 2017. Critical *Streptococcus suis* Virulence Factors: Are They All Really Critical? Trends Microbiol. Elsevier Ltd <https://doi.org/10.1016/j.tim.2017.02.005>.

Liu Z, Zheng H, Gottschalk M, Bai X, Lan R, Ji S, Liu H, Xu J. 2013. Development of Multiplex PCR Assays for the Identification of the 33 Serotypes of *Streptococcus suis*. PLoS One 8:e72070.

Jessing SG, Angen Ø, Inzana TJ. 2003. Evaluation of a multiplex PCR test for simultaneous identification and serotyping of *Actinobacillus pleuropneumoniae* serotypes 2, 5, and 6. J Clin Microbiol 41:4095–4100.

Schuwert L, Hoeltig D, Waldmann KH, Strutzberg-Minder K, Valentin-Weigand P, Rohde J. 2020. Serotyping and pathotyping of *Glaesserella parasuis* isolated 2012–2019 in Germany comparing different PCR-based methods. Vet Res 51:1–14.

Pai R, Gertz RE, Beall B. 2006. Sequential multiplex PCR approach for determining capsular serotypes of *Streptococcus pneumoniae* isolates. J Clin Microbiol 44:124–131.

Díaz-Zúñiga, J., More, J., Melgar-Rodríguez, S., Jiménez-Unión, M., Villalobos-Orchard, F., Muñoz-Manríquez, C., ... Paula-Lima, A. (2020). Alzheimer's Disease-Like Pathology Triggered by *Porphyromonas gingivalis* in Wild Type Rats Is Serotype Dependent. Frontiers in Immunology, 11(November), 1–16.

Migale R, Herbert BR, Lee YS, Sykes L, Waddington SN, Peebles D, Hagberg H, Johnson MR, Bennett PR, MacIntyre DA. 2015. Specific *Lipopolysaccharide* Serotypes Induce Differential Maternal and Neonatal Inflammatory Responses in a Murine Model of Preterm Labor. Am J Pathol 185:2390–2401

2. The authors attempted to work around this problem with DAPC but it needs to be further explained how the accessory genome was filtered into 1060 VRDGs either in the results or methods. Alternative to the DAPC method, authors could validate their phenotype associated orthologs with a more robust method other than selection of 100 validated principal

components for DAPC.

Response: We are so sorry that the description of this part is not clear enough. Discriminant Analysis of Principle Components (DAPC) was used in our study to determine whether the genotypes of the isolates (accessory genome) were distinct between virulent and non-virulent strains. The 100 validated principal components were determined by cross-validation, which accounted for approximately 82.73% of the total genetic variability (We sincerely apologize for our mistake that the original manuscript is wrongly written as “18%”, “82.73%” were calculated from the following figure, and “18%” has been corrected to “82.73%” in the line 384 of revised manuscript). Then, we filtered 1060 VRDGs with Fisher’s exact test, which were significantly distributed in virulent genomes.

DAPC is a core technique used in many population genetic studies (Miller et al., 2020). For example, Lucas et al. used DAPC to identify differences in the accessory genome of rhizosphere and bulk soil subpopulations; Thomas *et al.* used DAPC to identify genetic differences between disease-associated isolates and non-disease-associated isolates.

In view of the comment, we have modified the text accordingly (lines 138-147 and lines 335 -342):

Lines 138-147: “To obtain more valuable phylogenetic profiles, PPNet identifies virulence-related differential genes (VRDGs) by comparing the distribution of genes from virulent and non-virulent genomes (Fig. 2a). Each gene receives its own null hypothesis of no association to virulence, and a Fisher’s test is performed (see Method). VRDGs are defined as gene families that are overrepresented in virulent genomes. A total of 1060 VRDGs were identified, and phylogenetic profiles of VRDGs were used to infer an association network (Table S2). Figure 2d shows that VRDGs were predominantly present in virulent genomes when compared with non-virulent genomes. Finally, PPNet generated a total of 81 virulence-related gene association networks based on 81 binary similarity and dissimilarity measures (32, 33) (Table S3)”

Lines 335-342: “In order to obtain a more valuable phylogenetic profile for network prediction, PPNet requires strain grouping information and creates a 2×2 contingency table, the levels being presence or absence for the trait and

gene, respectively, with counts of the number of isolates in each cell. For each gene, we assume the null hypothesis i.e. it is independent of virulence, and uses Fisher's exact test to compute p-values. Finally, p-values were corrected by the False Discovery Rate. Genes with adjusted p value (p_{adj}) < 0.05 are considered phenotype-related differential genes, and the phylogenetic profile of these differential genes was used for network inference."

1. Lopes LD, Pereira e Silva M de C, Weisberg AJ, Davis EW, Yan Q, Varize C de S, Wu CF, Chang JH, Loper JE, Andreote FD. 2018. Genome variations between rhizosphere and bulk soil ecotypes of a *Pseudomonas koreensis* population. *Environ Microbiol* 20:4401–4414.
2. Wileman TM, Weinert LA, Howell KJ, Wang J, Peters SE, Williamson SM, Wells JM, Langford PR, Rycroft AN, Wren BW, Maskell DJ, Tucker AW. 2019. Pathotyping the Zoonotic Pathogen *Streptococcus suis*: Novel genetic markers to differentiate invasive disease-associated isolates from non-disease-associated isolates from England and Wales. *J Clin Microbiol* 57.
3. Miller JM, Cullingham CI, Peery RM. 2020. The influence of a priori grouping on inference of genetic clusters: simulation study and literature review of the DAPC method. *Heredity (Edinb)* 125:269–280.

2. Authors have used the tool Roary for ortholog determination, but gene presence/absence microbial GWAS can be conducted using Scoary (or others), another tool by the same group for statistically significant phenotype associated separation of ortholog data. Since this is an already established concept in the phenotype association field, these tools should be used or compared in relation to PPNet, and any advantages or disadvantages of both methods should be highlighted.

Response: According to the reviewer's suggestion, we have compared Scoary and our approach in the discussion section (lines 243-255):

Lines 243-255: "Another programme for determining genes associated with phenotypes is Kover, a k-mer-based software using machine learning algorithms which allows users to find some k-mers (sequences of k length) that are associated with phenotype (41). Kover recognises k-mer presence/absence rather than gene presence/absence, and is convenient to test other types of representations for genomic variants, such as single nucleotide polymorphisms (SNPs) and unitig level. However, Kover is less user friendly as users need to further annotate through sequence alignment to identify the cognate gene. Our approach and Scoary (42) both use Fisher's exact test to compare the significance of the presence/absence of genes associated with the different phenotypes, although do not detect SNPs in the orthologs of accessory genes because they are classified according to sequence similarity. In addition, Scoary considers the effect of population structure on gene distribution. However, Scoary is considered stringent, resulting in too few predictions (43), is not conducive to identifying subsequent network inference studies based on phylogenetic profiles."

3. The paper doesn't discuss or address the possibility of an ortholog of accessory genes having minor differences in sequence which associate with one phenotype over another or affect functional association in one group

over the other. It is worth discussing this, as other tools operate at the SNP and unitig level to identify such variants. PPNet would likely not detect these differentially present gene networks as they would not separate out at the ortholog level.

Response: Response: We appreciate your suggestion and have added some discussion to the revised manuscript (lines 243-255), where we have introduced and compared the advantages and disadvantages of PPNet with other methods i.e. Kover and Scoary

Lines 243-255: "Another programme for determining genes associated with phenotypes is Kover, a k-mer-based software using machine learning algorithms which allows users to find some k-mers (sequences of k length) that are associated with phenotype (41). Kover recognises k-mer presence/absence rather than gene presence/absence, and is convenient to test other types of representations for genomic variants, such as single nucleotide polymorphisms (SNPs) and unitig level. However, Kover is less user friendly as users need to further annotate through sequence alignment to identify the cognate gene. Our approach and Scoary (42) both use Fisher's exact test to compare the significance of the presence/absence of genes associated with the different phenotypes, although do not detect SNPs in the orthologs of accessory genes because they are classified according to sequence similarity. In addition, Scoary considers the effect of population structure on gene distribution. However,

Scoary is considered stringent, resulting in too few predictions (43), is not conducive to identifying subsequent network inference studies based on phylogenetic profiles.”

4. While PPNet is simple to download and install as the paper suggests. Once installed it seems to be missing a key python script (average_nucleotide_identity.py) rendering it non-functional. Authors should fix the github version and attempt installation and execution of PPNet on an independent system.

Response: We sincerely apologize for our mistake. Thanks for your valuable comments. PPNet requires users to install pyani module first, which includes the “average_nucleotide_identity.py” script. We have added this description in the “README.md” file of the github and installed and executed PPNet on an independent system.

5. The data presented in Figure 3 for choice of similarity/distance measure (Eqn. 60) doesn't match the data referenced in the Supplementary Table 2. In the table, AUROC and AUPR values exceed 100% and this has not been explained nor do the values always correspond to the ones in the figure. The chosen equation (C1 Eqn60) for the network construction is the only one with an AUROC >100% listed in the table.

Response: We sincerely apologize for our mistake. Thanks for your valuable

comments. You are right that what was recorded in the previously submitted Supplementary Table 2 were values of $-\log_{10}p_{\text{ROC}}$ and $-\log_{10}p_{\text{PR}}$. We have corrected the mistake in Supplementary Table 3.

6. Input datasets of binary accessory genomes (used in DAPC and network construction) could be provided as a supplemental table for review. This could also be of benefit to the people who study *S. suis* as a database of *S. suis* accessory orthologs.

Response: We appreciate the reviewer's suggestion. According to the reviewer's suggestion, the datasets are now provided in Supplementary Table 2.

Minor Comments

Figure Legend 1

- ANI and ANC thresholds are described as user-adjustable but the current version of PPNet appears to be hard coded.

Response: We have added user-adjustable parameters (-t1, -t2) -to PPNet for setting ANI and ANC thresholds, respectively.

- The figure clearly depicts what is meant by the term "phylogenetic profile". However, this should be better explained in the text as just a binary matrix as "phylogenetic profile" is a bit vague and could be referring to other things like phylogenetic tree topology.

Response: We have added an explanation of the phylogenetic profile in the legend of Figure 1 (lines 693-705):

“Figure 1 Schematic representation of the PPNet workflow. Taking genomes data and grouping information of strains as the input, each genome data goes through a set of filtration steps, including removal of poor-quality genomes based on N50 and removal of duplicate genomes based on ANI and ANC, the thresholds for each step set by the user. Next, the obtained high-quality genomes are automatically annotated and a preliminary phylogenetic profile is constructed. The phylogenetic profile is represented by a binary matrix, each row represents an ortholog, each column represents a strain and the 1 or 0 in each row refers to the presence or absence of the ortholog in each strain, respectively. The preliminary phylogenetic profile is then filtered by fisher’s exact test, only the phylogenetic profile of orthologs with significant differences in distribution across strain groups is retained. Finally, the association coefficients among the orthologs are calculated based on the similarity of their phylogenetic profiles, these results are saved as the output and can be imported to Cytoscape for visualization.”

Figure Legend 2

-Panel C: It needs to be mentioned that the tree is constructed using the accessory ortholog profile in the legend as it is mentioned in the text. This could be misinterpreted as a whole genome tree.

Response: We agree with the reviewer's remark. According to your suggestion, we have rewritten the legend of Figure 2c (lines 711-713):

“A phylogenetic tree of 551 *S. suis* strains based on the binary presence and absence of accessory genes. Outer ring provides information about the virulent (red) and non-virulent serotypes (blue), respectively.”

-Panel D has $-\log_{10}(p.adj)$ mentioned but it's not clear what these values reflect.

Response: Thank for your suggestion, we have rewritten the legend of Figure 2d (lines 714-721):

“Heatmap visualizing the distribution of VRDGs in the genomes of *S. suis*. The presence (dark blue areas) or absence (light blue areas) of 1060 VRDGS are shown in the heatmap. Each row indicates a VRDG and was clustered by hierarchical clustering based on VRDG distribution. Each column indicates a strain, which was divided into virulent group (red) and non-virulent group (blue). A colored strip from red to yellow shown on left side of the heatmap correspond to $-\log_{10}(p.adj)$ values from low to high. The $p.adj$ values are the adjusted p values computed by Fisher's exact test under the null hypothesis that the presence/absence of this gene is unrelated to virulence and adjusted by False Discovery Rate.”

Figure Legend 3

-Panel a cites Supplemental Table 5 when it should be Supplemental Table 2

Response: We sincerely apologize for our mistake. We have corrected this mistake (lines 722-723):

“Figure 3 Performance of network inference methods. (a) Assessment of network inference methods listed in Supplementary Table 3.”

-Panel b highlights the relevance of Eqn. 60 for S suis but the colored boxes are not explained.

Response: Thank for your suggestion, we have rewritten the legend of Figure 3b (lines 728-733) and we have added Supplementary Figure 1 to illustrate this part:

“The scatter plot depicts the minimum distance vs. AUROC by 81 binary similarity and distance measures. According to the distribution, the final 43 dots obtained from 81 equations were divided into four groups (G1, G2, G3 and G4) through hierarchical clustering (Fig. S1), represented by four different colored boxes. OCHIAI-II similarity (Eq. 60) present in the first group had a relatively short minimum distance and the second-highest AUROC value.”

Figure S1:

Figure S1 Dendrogram of the 81 binary similarity and distance measures. The 81 equations were divided into four groups by hierarchical clustering based on the minimum distance of the ROC curve to the theoretical optimum point and AUROC (red boxes).

Figure Legend 5

-Cite the Supplemental table listing each "Mx" number and associated genes in the legend.

Response: Thanks for your valuable comments. According to your suggestion, we have added the description in the legend of Figure 5 (lines 743-744):

“The “Mx” in the figure represents module x in Supplementary Figure 2.”

Line 38 - typo: experimental -> experiments

Response: According to your suggestion, “experimental” has been corrected to “experiments” in the revised manuscript (line 31):

“Selected network associations were validated experimentally by bacterial two-hybrid experiments.”

Lines 44-47 - same as the abstract

Response: We have rewritten this part (lines 35-42):

“This study developed PPNet, the first tool that can be used to infer large-scale bacterial functional association networks of a single species. PPNet includes a method of assign the uniqueness of a bacterial strain by the values of Average Nucleotide Identity (ANI), Average Nucleotide Coverage (ANC). PPNet collected 81 binary similarity and distance measures for phylogenetic profiling, evaluated and divided them into four groups. PPNet can effectively capture gene networks that are functionally related to phenotype from publicly prokaryotic genomes, as well as provides valuable results for downstream analysis and experiment testing.”

Line 128 - typo: publically ->publicly

Response: “publically” has been corrected to “publicly” in the revised manuscript (line 112):

“To demonstrate its usefulness, PPNet was used to infer the virulence-related gene association network of *S. suis* from publicly available data.”

Line 156-157 - significantly distributed is a misnomer as no direct gene presence-absence statistics were present. Perhaps "overrepresented in virulent genomes"

Response: Thanks for your valuable comments. According to your suggestion, “significantly distributed” has been corrected to “overrepresented” in the revised manuscript (lines 141-142):

“VRDGs are defined as gene families that are overrepresented in virulent genomes.”

December 1, 2022

Dr. Hui Jin
Huazhong Agricultural University
State Key Laboratory of Agricultural Microbiology, College of Veterinary Medicine, Huazhong Agricultural University
Wuhan
China

Re: Spectrum03871-22R1 (PPNet: Identifying functional association networks by phylogenetic profiling of prokaryotic genomes)

Dear Dr. Hui Jin:

Your manuscript has been accepted, and I am forwarding it to the ASM Journals Department for publication. You will be notified when your proofs are ready to be viewed.

Sincerely,

Sébastien Faucher
Editor, Microbiology Spectrum

Journals Department
Supplemental file2: Accept
Supplemental file1: Accept